

# Should we correct the bias in Confidence Bands for Repeated Functional Data?

**COMPUTO**

ISSN 2824-7795

Emilie Devijver[1]    CNRS, Univ. Grenoble Alpes, Grenoble INP, LIG, 38000 Grenoble, France

Adeline Samson    Univ. Grenoble Alpes, CNRS, Grenoble INP, LJK, 38000 Grenoble, France

Date published: 2025-10-29    Last modified: 2025-10-29

### Abstract

While confidence intervals for finite quantities are well-established, constructing confidence bands for objects of infinite dimension, such as functions, poses challenges. In this paper, we explore the concept of parametric confidence bands for functional data with an orthonormal basis. Specifically, we revisit the method proposed by Sun and Loader, which yields confidence bands for the projection of the regression function in a fixed-dimensional space. This approach can introduce bias in the confidence bands when the dimension of the basis is misspecified. Leveraging this insight, we introduce a corrected, unbiased confidence band. Surprisingly, our corrected band tends to be wider than what a naive approach would suggest. To address this, we propose a model selection criterion that allows for data-driven estimation of the basis dimension. The bias is then automatically corrected after dimension selection. We illustrate these strategies using an extensive simulation study. We conclude with an application to real data.

*Keywords:* functional data, repeated data, confidence band, Kac-Rice formulae, bias, dimension selection

# Contents

---

[1]Corresponding author: emilie.devijver@univ-grenoble-alpes.fr

# 1   Introduction

Functional data analysis is widely used for handling complex data with smooth shapes, finding applications in diverse fields such as neuroscience (e.g., EEG data, Zhang (2020)), psychology (e.g., mouse-tracking data, Quinton et al. (2017)), and sensor data from daily-life activities (Jacques and Samardžić (2022)). It consists in estimating a function, which is an object of infinite dimension. It is important to accompany the function estimate with a measure of uncertainty, for example through a simultaneous confidence band. This task presents several challenges: the confidence band must control the simultaneous functional type-I error rate, as opposed to point-wise rates; it must strike a balance between being sufficiently conservative to maintain a confidence level while not being overly so as to render it meaningless; and the method used to construct this confidence band should be computationally feasible for practical application (Ramsay (2005)).

We consider several independent observations of the same function, i.e., noisy functional data. To analyze this noisy data, a classic approach consists in either averaging pointwise the data, applying a kernel function to smooth the noise or projecting the data onto a functional space defined by a family of functions (Kokoszka and Reimherr (2017) Chapter 3, Li, Qiu, and Xu (2022)). The pointwise empirical mean is not a function but a vector of estimated discrete points. On the other hand, with the two other approaches, the targeted function is then an approximation of the true function and obtained confidence bands are thus bands of this approximated function. In such context, several methods have been proposed to construct a confidence band that controls the simultaneous functional type-I error rate. In the case of a single individual (no repetition) but with many time points, some methods study the asymptotic distribution of the infinity norm between the targeted approximated function and its estimator, the asymptotics being in the number of time points (Hall (1991), Claeskens and Van Keilegom (2003)). These approaches only work for large datasets in time and are likely to be too conservative otherwise. For small samples, bootstrap methods have been developed to compute the confidence band (Neumann and Polzehl (1998), Claeskens and Van Keilegom (2003)), but with a high computational cost. Another approach consists in constructing confidence bands based on the volume-of-tube formula. Sun and Loader (1994) studied the tail probabilities of suprema of Gaussian random processes. This approach is based on an unbiased linear

estimator of the regression function, which corresponds to a band of the approximated targeted function. Zhou, Shen, and Wolfe (1998) used the volume-of-tube formula for estimation by regression splines. Krivobokova, Kneib, and Claeskens (2010) applied this method for the construction of confidence bands using penalized spline estimators. Bunea, Ivanescu, and Wegkamp (2011) propose a threshold-type estimator and derive error bounds and simultaneous confidence bands. In the case of several observations of the same function, Liebl and Reimherr (2023) proposed a method based on random field theory and the volume-of-tube formula, leveraging the Kac-Rice formula. Their approach introduces a quantile that varies with location, which allows to achieve their fairness property. Their confidence band uses an unbiased estimator. Unlike other methods, it does not require estimating the full covariance function of the estimator, but only its diagonal, which reduces computational time. From a practical viewpoint, Sachs, Brand, and Gabriel (2022) introduced a package to popularize simultaneous confidence bands in the context of survival analysis. Telschow and Schwartzman (2022) propose a simultaneous confidence band based on the Gaussian kinematic formula. Again, it assumes access to an asymptotically unbiased estimator of the function of interest. The coverage is thus guaranteed in the asymptotic setting after removing the bias and by targeting the approximated smoothed function. Their paper considers the non-gaussian and non-stationary cases. Wang (2022) proposed a simultaneous Kolmogorov-Smirnov confidence band by modeling the error distribution, thus avoiding the estimation of the covariance structure of the underlying stochastic process. They rely on B-splines for the estimation of the mean curve. Note that recent extensions have been proposed in Telschow et al. (2023) to construct simultaneous confidence bands, or based on conformal prediction in Diquigiovanni, Fontana, and Vantini (2022), or having a prediction goal in mind in Hernández, Cugliari, and Jacques (2024) by considering functional time series data. These different papers use approximation of the function of interest and do not deal with the non asymptotic associated bias.

We want to work in the non asymptotic setting and to propose confidence band of the original true function, not only on the approximated one (obtained by smoothing kernel or by projection). We do not work with the empirical mean estimator for two main reasons. It does not inherit functional properties, especially its regularity. Furthermore, the dimension of the empirical mean estimator is larger than the dimension of a projection or kernel estimator and this induces a larger variance and wider confidence band. Thus we focus on functional estimator.

The difference between the original function and its approximation is a functional (deterministic) bias that depends on the quality of the kernel or the projection. This bias may be neglected in the asymptotic setting where the number of observations goes to infinity, but not with finite sample sizes. It has to be taken into account in the construction of the confidence band. Both methods (kernel and projection) rely on the choice of an hyperparameter, the kernel bandwidth in the first case and the dimension of the functional projection basis in the second. The choice of this hyperparameter is crucial to control the type-I-error rate of the confidence band viewed as a band of the true function in a non-asymptotic setting. This is the question we explore in this paper.

Sun and Loader (1994) proposed a bias correction for a particular class of functions but left the smoothing parameter choice open, leading to an unusable estimator. In the nonparametric framework, the bias is approximated using the estimator of the second derivative of the underlying mean function (Xia (1998)). But in general, there is a lack of discussion on how to handle the bias of the functional estimator, even in the simple case of a functional space of finite dimension. Hard-thresholding approaches, cross-validation methods (Li, Qiu, and Xu (2022)) or model selection framework could be used to select the best dimension. However, these approaches need to be adapted to the specific case of controlling the level of a confidence band. Few references exist on this subject. For example, while the model selection paradigm has been extensively studied in the literature, in multivariate statistics or functional data analysis (e.g., Goepp, Bouaziz, and Nuel (2025), Aneiros, Novo, and Vieu

(2022), Basna, Nassar, and Podgórski (2022)), it has not been explored in the context of confidence band construction.

The objective of this paper is to address the bias problem in confidence band construction for a general function, in the non-asymptotic setting with respect to the number of individuals and the number of time points. We adopt the point of view of projecting data onto a functional space, because the functional bias is easier to study and control, than when smoothing data with a kernel. Especially, when the family is an orthonormal basis, e.g., the Legendre basis (with the standard scalar product) or Fourier (with another scalar product), the projection is explicit and it is possible to obtain theoretical results. Moreover, the functional space offers a key advantage: it reduces the problem of inference to the estimation of coefficients, for example by least squares or maximum likelihood estimation. The function estimator is then simply an average after projection onto the functional base (Ramsay (2005)). Our contributions are as follows:

- we disentangle the bias issue by explicitly defining the parameter of interest within the approaches of Sun and Loader (1994) and Telschow et al. (2023);
- we propose a confidence band for the true function of interest, including a bias correction. This provides a collection of debiased confidence bands. We also propose a criteria to select the best band, by splitting the sample into two sub-samples;
- we propose a second heuristic method for selecting the dimension of the approximation space, treating it as a model selection problem, with a trade-off between conservatism and confidence level assurance; this approach does not correct the bias of each band of the collection but selects a band with a negligible bias;
- we illustrate the proposed strategies and compare them to cross-validation or threshold approaches;
- we also illustrate the impact of the choice of the functional family, including non-orthonormal families.

The paper is organized as follows: Section 2 introduces the functional regression model, the considered functional family and the corresponding approximate regression models, as well as an estimator defined in the finite space, along with descriptions of the error terms. In Section 3, we propose a confidence band for the approximate regression function in the space of finite dimension, where the dimension is fixed. Section 4 proposes a strategy to construct a confidence band for the true function. This last confidence band being too conservative, Section 5 introduces a model selection criterion to select the best confidence band, doing a trade-off between conservatism and confidence level assurance. Section 6 illustrates the different estimating procedures of the confidence band. Section 7 proposes an application on real data. Section 8 ends the paper by a conclusion and discussion of perspectives.

## 2  Statistical Model

In this paper, we consider time series as discrete measurements of functional curves. We first present the general functional regression model (Section 2.1) where the regression function belongs to a finite functional family of dimension $L^*$. In practice, this dimension $L^*$ is unknown and we will work on functional space of dimension $L$. The regression model on the finite family of functions is presented in Section 2.2, and an estimator is proposed in Section 2.3, with a description of the error terms.

In the rest of the paper, we consider the space $L^2([0, 1])$ with its standard scalar product $< f_1, f_2 >= \int_0^1 |f_1(t)f_2(t)|dt$, for $f_1, f_2 \in L^2([0, 1])$. The notation *Vect* denotes the linear span.

## 2.1 Functional regression model

Let $y_{ij}$ be the measure at fixed time $t_j \in [a, b]$ for individual $i = 1, \ldots, N$, with $j = 1, \ldots, n$. The case with time observations dependent of the individuals is a natural extension of this case, but is not treated in this paper. We restrict ourselves to $[a, b] = [0, 1]$, without loss of generality. We assume these observations are discrete-time measurements of individual curves, which are independent and noisy realisations of a common function $f$ that belongs to a functional space. Thus for each individual $i$, we consider the following functional regression model

$$y_{ij} = f(t_j) + \varepsilon_{ij},$$

where $\varepsilon_{i.} = (\varepsilon_{i1}, \ldots, \varepsilon_{in})$ is the measurement noise assuming that the $\varepsilon_i$ are independent.

For each individual $i = 1, \ldots, N$, we denote $y_{i.} = (y_{i1}, \ldots, y_{in})$, $t_. = (t_1, \ldots, t_n)$ and $f(t_.) = (f(t_1), \ldots, f(t_n))$ the $n \times 1$ vectors of the observations, times and function $f$ evaluated in $t_.$, respectively. We also denote $\mathbf{y} = (y_{1.}, \ldots, y_{N.})$ the whole matrix of observations.

The unknown regression function $f$ lives in an infinite space and can not be directly estimated. First we reduce the dimension by projecting $f$ on a function space. Let us consider $L^*$ functions $(B_\ell^{L^*})_{1 \leq \ell \leq L^*}$ assumed to be linearly independent and the corresponding functional space $\mathcal{S}^{L^*} = Vect((t \mapsto B_\ell^{L^*}(t))_{1 \leq \ell \leq L^*})$. Then, for any $f \in \mathcal{S}^{L^*}$, there exists a unique vector of coefficients $(\mu_\ell^{L^*})_{1 \leq \ell \leq L^*}$ such that, for all $t$, $f(t) = \sum_{\ell=1}^{L^*} \mu_\ell^{L^*} B_\ell^{L^*}(t)$. The regression function $f$ verifies the following assumption:

**Assumption 1.** The function $f$ belongs to the space $\mathcal{S}^{L^*}$ of dimension $L^*$. It is denoted $f^{L^*}$ and defined as:

$$f(t) = f^{L^*}(t) = \sum_{\ell=1}^{L^*} \mu_\ell^{L^*} B_\ell^{L^*}(t).$$

Many functional spaces are available in the literature, as Splines, Fourier or Legendre families. We introduce the following assumption:

**Assumption 2.** The functional family $(t \mapsto B_\ell^{L^*}(t))_{1 \leq \ell \leq L^*}$ is orthonormal with respect to the standard scalar product $< ., . >$.

Note that if Assumption 2 holds, one get $\mu_\ell^{L^*} = < f^{L^*}, B_\ell^{L^*} >$ for $\ell = 1, \ldots, L^*$. The Legendre family is orthonormal, the Fourier family is orthogonal for the standard scalar product (but not orthonormal), and the B-splines family is not orthogonal. We will illustrate the impact of using one family or the other in Section 6.6.

We also consider a functional noise through the following assumption.

**Assumption 3.** The sequence $\varepsilon_i$ is functional and allows Karhunen-Loève $L^2$ representation: there exists a sequence of coefficients $(c_{i\ell})_{1 \leq \ell}$ such that

$$\varepsilon_{ij} = \sum_{\ell \geq 1} c_{i\ell} \phi_\ell(t_j),$$

where the functions $(\phi_\ell)_{1 \leq L}$ can be written through eigenvalues and eigenfunctions of the covariance matrix $cov(\varepsilon_{ij}, \varepsilon_{ij'})$. Practically, we assume this sum to be finite, as done for example in Chen and Song (2015): there exists $L^\varepsilon$ such that

$$\varepsilon_{ij} = \sum_{1 \leq L \leq L^\varepsilon} c_{i\ell} \phi_\ell(t_j).$$

We also assume that the coefficients are Gaussian: for all $i = 1, \ldots, N$ and $\ell = 1, \ldots, L^\varepsilon$,

$$c_{i\ell} \sim_{iid} \mathcal{N}(0, \sigma^2).$$

Note that we could also work with elliptic distributions instead of the Gaussian distribution. The results of the paper would be the same but require more technical details. For simplicity, we focus on the Gaussian case.

Assumption 1 and Assumption 3 imply that each curve $y_i$ belongs to a finite family: for $j = 1, \dots, n$,

$$y_{ij} = \sum_{\ell=1}^{L^*} \mu_\ell^{L^*} B_\ell^{L^*}(t_j) + \sum_{\ell=1}^{L^\varepsilon} c_{i\ell} \phi_\ell(t_j).$$

As the observations are recorded at discrete time points $(t_j)_{1 \le j \le n}$, for $L \in \mathbb{N}$, let us denote $\mathbf{B}^L$ the matrix of $n \times L$ with coefficient in row $j$ and column $\ell$ equal to $B_\ell^L(t_j)$, and the basis for the noise $\Phi^{L^\varepsilon} = (\phi_\ell(t_j))_{1 \le \ell \le L^\varepsilon, 1 \le j \le n}$. Let us introduce $c_{i\cdot} = (c_{i1}, \dots, c_{iL^\varepsilon})$ the $L^\varepsilon \times 1$ vector. Then $\varepsilon_{i\cdot} = \Phi^{L^\varepsilon} c_{i\cdot}$. The vectors $y_{i\cdot} \in \mathbb{R}^n$ are thus independent and $y_i \sim \mathcal{N}_n(f(t_\cdot), \sigma^2 \Sigma^{L^\varepsilon})$ with $\Sigma^{L^\varepsilon} = \Phi^{L^\varepsilon}(\Phi^{L^\varepsilon})^T$.

The objective of this paper is to construct a tight confidence bound for $f^{L^*}$ using data $(y_{ij})_{ij}$. The main challenge is that the true dimension $L^*$ is unknown. In practice, we can only work with a projected version on the space $\mathcal{S}^L$ with $L \in \{L_{\min}, \dots, L_{\max}\}$. Two issues are introduced: the bias induced by this projection and the choice of $L$. In this paper, we treat both of them in a non asymptotic setting to construct confidence band with a control level.

In the rest of the paper, we will work with a collection of models defined by the dimension $L$ with $L \in \{L_{\min}, \dots, L_{\max}\}$, $L_{\max}$ being chosen to be sufficiently large the user, expecting that $L^* \le L_{\max}$. $L_{\max}$ has to be large enough to do overfitting. We will study the functional bias and its asymptotic behavior. The we will propose different strategies to choose the best dimension among the different collections.

First, in Section 2.2 and Section 2.3, we define for a fixed $L$ the corresponding regression model and its estimator. Then Section 3, Section 4 and Section 5 will introduce the different bandwidths.

## 2.2 Approximation of the model on a finite family

Let $f^{L^*} \in \mathcal{S}^{L^*}$ with $L^*$ unknown, and consider the space $\mathcal{S}^L$ for $L$ fixed in $\{L_{\min}, \dots, L_{\max}\}$. As $\mathcal{S}^L$ is the linear span of linearly independent functions, there is always a unique vector $\mu^L$ of coefficients defining $f^L(t) = \sum_{\ell=1}^L \mu_\ell^L B_\ell^L(t) = B^L(t)\mu^L$ such that

$$f^L = \arg\min_{f \in \mathcal{S}^L} \{\|f^{L^*} - f\|_2^2\}.$$

If the family is orthonormal, it corresponds to the projected coefficients $\mu_\ell^L$:

$$\mu_\ell^L := <f^{L^*}, B_\ell^L>.$$

We know that under Assumption 1, $f^{L^*, L^*} = f^{L^*}$. Moreover under Assumption 2,

$$\mu_\ell^L = \mu_\ell^{L^*} \quad for\ \ell = 1, \dots, \min(L, L^*).$$

It is interesting to note that this is not true when the basis is not orthonormal.

In practice, data are observed at discrete time, we consider the operator $\mathbf{P}^L$ defined as the matrix $\mathbf{P}^L = ((\mathbf{B}^L)^T \mathbf{B}^L)^{-1} (\mathbf{B}^L)^T$ of size $L \times n$. This coincides with the orthogonal projection when we deal with an orthonormal basis, but this formula also holds for non orthonormal family, coming back to

the least square estimator on a specified family. Then we define the coefficients $\underline{\mu}^L$ which are the coefficients of $\mu^L$ approximated on the vector space, denoted $\mathbf{S}^L$, defined by the matrix $\mathbf{B}^L$.

$$\underline{\mu}^L := \mathbf{P}^L \mathbf{B}^{L^*} \mu^{L^*}.$$

Note that

- When $L \geq L^*$, when $n > L$, we have for $\ell = 1, \ldots, L^*$,

$$\underline{\mu}_\ell^L = \mu_\ell^{L^*}.$$

- When $L < L^*$, for $\ell = 1, \ldots, L$,

$$\underline{\mu}_\ell^L \neq \mu_\ell^{L^*}.$$

The corresponding finite approximated regression function is denoted $\underline{f}^L$ and is defined, for all $t \in [0, 1]$, as

$$\underline{f}^L(t) = B^L(t)\underline{\mu}^L.$$

We can observe the following, linking $L, L^*$ and the number of timepoints $n$: under Assumption 1 and Assumption 2, the diagonal elements of $\mathbf{P}^L\mathbf{B}^{L^*}$ are such that for $\ell = 1, \ldots, \min(L, L^*)$, $[\mathbf{P}^L\mathbf{B}^{L^*}]_{\ell\ell} = 1$. Moreover,

- When $n \to \infty$, for $\ell = 1, \ldots, L$

$$\underline{\mu}_\ell^L \to \mu_\ell^{L^*}.$$

- If $n > L^*$, then $f^{L^*} = f^{L^*, L^*} = \underline{f}^{L^*, L^*}$.

The last point is particularly interesting, relating $n$ the number of timepoints to the true level $L^*$ such that the discretized functions correspond to the true one.

## 2.3 Estimator

Let $L \in \{L_{\min}, \ldots, L_{\max}\}$. This section presents the least square estimator of the regression function on the space of dimension $L$ defined by the family $\mathbf{B}^L$ and discusses its error, i.e. its bias and its behavior with respect to $L$ and $n$.

### 2.3.1 Estimation of the regression function

When considering the estimation of the regression function $f^{L^*}$ on the space of dimension $L$ defined by the family $\mathbf{B}^L$, we do not directly estimate $f^{L^*}$ but its projection on this finite space, which corresponds to the projected function $\underline{f}^L(t)$ and its associated coefficients $(\underline{\mu}_\ell^L)_{1 \leq \ell \leq L}$.

**Definition 2.1.** The vector of coefficients $(\underline{\mu}_\ell^L)_{1 \leq \ell \leq L}$ is estimated by the least square estimator $\hat{\underline{\mu}}^L$ defined as:

$$\hat{\underline{\mu}}^L := \frac{1}{N}\sum_{i=1}^{N} \mathbf{P}^L y_{i.}.$$

For a fixed $t \in [0, 1]$, the estimator of the function $\underline{f}^L(t)$ is defined by:

$$\hat{\underline{f}}^L(t) = \sum_{\ell=1}^{L} \hat{\underline{\mu}}_\ell^L B_\ell^L(t) = B^L(t)\hat{\underline{\mu}}^L. \tag{1}$$

Equation 1 directly implies that the estimator is thus the empirical mean of the functional approximation of each individual vector of observations. Because we work with least squares estimators, we can easily study the error of estimation of $\hat{\underline{\mu}}^L$ and $\hat{\underline{f}}^L$.

**Proposition 2.1.** *Under Assumption 1 and Assumption 3, we have*

$$\hat{\underline{\mu}}^L \sim \mathcal{N}_L\left(\underline{\mu}^L, \frac{\sigma^2}{N}\Sigma_B^{L,L^\varepsilon}\right),$$

*where the $L \times L$ covariance matrix $\Sigma_B^{L,L^\varepsilon}$ is defined as $\Sigma_B^{L,L^\varepsilon} := \mathbf{P}^L \Sigma^{L^\varepsilon}(\mathbf{P}^L)^T$ with $\Sigma^{L^\varepsilon} = \Phi^{L^\varepsilon}(\Phi^{L^\varepsilon})^T$.*

*Moreover, $B^L \mathbf{P}^L y_i$ is a Gaussian process with mean $\underline{f}^L$ and covariance function $(s,t) \mapsto \sigma^2 B^L(s)\Sigma_B^{L,L^\varepsilon}(B^L(t))^T$, and $(\hat{\underline{f}}^L - \underline{f}^L)$ is a centered Gaussian process with covariance function $C^L : (s,t) \mapsto \frac{\sigma^2}{N} B^L(s)\Sigma_B^{L,L^\varepsilon} B^L(t)^T$.*

The proof is given in Appendix.

Even if the estimator $\hat{\underline{f}}^L$ is defined on the functional space associated to $\mathbf{S}^L$, our approach consists in seeing it as an estimator of the function $f^{L^*}$ which lies in the space $\mathcal{S}^{L^*}$. In that case, the error includes a functional approximation term due to the approximation of $f^{L^*}$ on the space $\mathcal{S}^L$, which will be nonzero if $L \neq L^*$. It corresponds to the bias of the estimator $\hat{\underline{f}}^L$, i.e. the difference between its expectation and the true $f^{L^*}$. Indeed, recalling that $f^{L^*} = \underline{f}^{L^*,L^*}$, the error of estimation can be decomposed into

$$\hat{\underline{f}}^L(t) - f^{L^*}(t) = \hat{\underline{f}}^L(t) - \underline{f}^L(t) + \underline{f}^L(t) - \underline{f}^{L^*,L^*}(t) =: Stat_L(t) + Bias_L(t), \tag{2}$$

The first term $Stat_L(t) = \hat{\underline{f}}^L(t) - \underline{f}^L(t)$ is the (unrescaled) statistics of the model. The second term $Bias_L(t) = \mathbb{E}(\hat{\underline{f}}^L(t)) - \underline{f}^{L^*,L^*}(t)$ is the bias of the estimator $\hat{\underline{f}}^L(t)$ when estimating the true function $\underline{f}^{L^*,L^*}(t)$.

Let us remark that this bias is different than the bias of the estimator $\hat{\underline{f}}^L(t)$ when estimating the projected function $\underline{f}^L = f^{L^*}$, which is 0. The two terms defined in Equation 2 are more detailed in the two next subsections.

### 2.3.2 Statistics

The statistics of the model, $t \mapsto Stat_L(t) = \hat{\underline{f}}^L(t) - \underline{f}^L(t)$, is a random functional quantity which depends on the estimator $\hat{\underline{f}}^L$. From Proposition 2.1, for any $t \in [0,1]$, we define the centered and rescaled statistics $Z_L(t)$:

$$Z_L(t) := \frac{Stat_L(t)}{\sqrt{\mathrm{Var}(Stat_L(t))}} = \frac{\hat{\underline{f}}^L(t) - \underline{f}^L(t)}{\sqrt{C^L(t,t)}} \sim \mathcal{N}(0,1).$$

The covariance function can be naturally estimated using the observations $y_{i\cdot}$ as

$$\hat{C}^L(s,t) = \frac{1}{N-1}\sum_{i=1}^N (B^L(s)\mathbf{P}^L y_{i\cdot} - \hat{\underline{f}}^L(s))(B^L(t)\mathbf{P}^L y_{i\cdot} - \hat{\underline{f}}^L(t)).$$

 **2.3.3 Bias**

 The bias is due to the fact that the estimation is potentially performed in a different (finite) space
 than the space where the true function $f^{L^*, L^*}$ lives. This is a functional bias, which is not random. It
 corresponds to the approximation of $f^{L^*}$ from $\mathcal{S}^{L^*}$ to the space $\mathcal{S}^L$. It can be written as follows:

$$Bias_L(t) = B^L(t)\underline{\mu}^L - B^{L^*}(t)\mu^{L^*}.$$

 It depends on $L$ and on the sample size through $\underline{\mu}^L$. Let us describe its behavior. When $L < L^*$ and
 the family is orthonormal, the approximation is the orthogonal projection and we have that

$$Bias_L(t) = \sum_{\ell=1}^{L} B_\ell^L(t)\underline{\mu}_\ell^L - \sum_{\ell=1}^{L^*} B_\ell^{L^*}(t)\underline{\mu}_\ell^{L^*} = \sum_{\ell=L+1}^{L^*} B_\ell^{L^*}(t)\underline{\mu}_\ell^{L^*}.$$

 From Proposition 2.1, we can directly deduce the following proposition:

 **Proposition 2.2.** *Under Assumption 1 and Assumption 3, the bias is, for all $t \in [0, 1]$,*

 • *for $L < L^*$, $Bias_L(t) \neq 0$,*
 • *for $L \geq L^*$, $Bias_L(t) = 0$.*

 Note that the bias is not 0 even when $n \to \infty$, as soon as $L < L^*$. Thus, a correct selection of the
 dimension $L$ is an issue.

 In the next section, we explain how we use this property to derive confidence bands of $\underline{f}^L$ and $f^L$.

## 3 Preliminary step: Confidence Bands of $\underline{f}^L$ and $f^L$ for a fixed $L$

 In this Section, we present some well known results on constructing a confidence band of $\underline{f}^L$ and $f^L$.
 As its definition and properties are essential to construct the next steps about confidence band of
 $f^{L^*}$, we have decided to recall them in details.

 The objective is to construct a confidence band for the two functions $\underline{f}^L$ and $f^L$, based on the
 observations **y**, for a given value $L \in \{L_{\min}, \dots, L_{\max}\}$. The band for $\underline{f}^L$ enters the framework
 proposed by Sun and Loader (1994) which relies on an unbiased and linear estimator of the function
 as the estimator $\hat{\underline{f}}^L$ is an unbiased estimator of $\underline{f}^L$. We recall in Section 3.1 the construction of this
 confidence band which attains a given confidence level in a non-asymptotic setting, that is for a finite
 number of observations $n$ for each individual. Then in Section 3.2, we prove that the confidence band
 proposed by Sun and Loader (1994) can be viewed as a confidence band for $f^L$ with an asymptotic
 confidence level, the asymptotic framework being considered when $n \to \infty$.

### 3.1 Confidence band for $\underline{f}^L$

 Consider $1 - \alpha$ a fixed confidence level. The aim is to find a function $d^L$ such that

$$\mathbb{P}\left(\forall t \in [0, 1], \ \hat{\underline{f}}^L(t) - d^L(t) \leq \underline{f}^L(t) \leq \hat{\underline{f}}^L(t) + d^L(t)\right) = 1 - \alpha.$$

 Consider the normalized statistics $Z_L(t)$ which is a centered and reduced Gaussian process. We want
 to find the quantile $q^L$ satisfying

$$q^L = \arg\min_q \left\{ \mathbb{P}\left( \max_{t \in [0,1]} |Z_L(t)| \le q \right) = 1 - \alpha \right\}. \tag{3}$$

Then we can take $d^L(t) = q^L \sqrt{C^L(t,t)}$. The covariance function $C^L(t,t)$ can be replaced by its estimator $\hat{C}^L(t,t)$, making the distribution a Student's distribution with $N-1$ degrees of freedom. Thus, it only requires to be able to compute the critical value $q^L$.

This can be done following Sun and Loader (1994) who propose a confidence band for a centered Gaussian process. Their procedure is based on an unbiased linear estimator of the function of interest, which is the case for $\hat{\underline{f}}^L$ when we consider a band for $\underline{f}^L$. We recall their result in the following proposition, the computation of the value $q^L$ is detailed thereafter. Note that the presentation of Telschow and Schwartzman (2022) is similar to the one adopted in this paper.

**Theorem 3.1** (Sun and Loader (1994)). *Set Assumption 1 and Assumption 3 and a probability $\alpha \in [0,1]$. Then, we have*

$$\mathbb{P}\left( \forall t \in [0,1], \left| \hat{\underline{f}}^L(t) - \underline{f}^L(t) \right| \le \hat{d}^L(t) \right) = 1 - \alpha$$

*with*

$$\hat{d}^L(t) = \hat{q}^L \sqrt{\hat{C}^L(t,t)/N}$$

*and $\hat{q}^L$ defined as the solution of the following equation, seen as a function of $q^L$:*

$$\alpha = \mathbb{P}\left( |t_{N-1}| > q^L \right) + \frac{\|\tau^L\|_1}{\pi} \left( 1 + \frac{(q^L)^2}{N-1} \right)^{-(N-1)/2}, \tag{4}$$

*with $(\tau^L)^2(t) = \partial_{12}c(t,t) = Var(Z_L'(t))$ where we denote $\partial_{12}c(t,t)$ the partial derivatives of a function $c(t,s)$ in the first and second coordinates and then evaluated at $t = s$.*

We can thus deduce a confidence band of level $1 - \alpha$ for $\underline{f}^L$:

$$CB_1(\underline{f}^L) = \left\{ \forall t \in [0,1], \left[ \hat{\underline{f}}^L(t) - \hat{d}^L(t); \hat{\underline{f}}^L(t) + \hat{d}^L(t) \right] \right\}.$$

The value $\hat{q}^L$ is defined implicitly in Equation 4 which involves the unknown and not explicit quantity $t \mapsto \tau^L(t)$. Liebl and Reimherr (2023) propose to estimate $\tau^L(t)$, for all $t$, by

$$\hat{\tau}^L(t) = \left( \widehat{Var}((U^L)_1'(t), \ldots, (U^L)_N'(t) \right)^{1/2}$$

$$= \left( \frac{1}{N-1} \sum_{i=1}^N \left( (U^L)_i'(t) - \frac{1}{N} \sum_{j=1}^N (U^L)_j'(t) \right)^2 \right)^{1/2},$$

where $U_i^L(t) = (P^L y_i.(t) - \hat{\underline{f}}^L(t))/(\hat{C}^L(t))^{1/2}$ and $(U^L)_i'$ is a smooth version of the differentiated function $U_i^L$. Then we take the $L_1$-norm of $\hat{\tau}^L$.

Let us describe the behavior of $\hat{d}^L$:

- $\|\hat{d}^L\|_\infty$ increases with $L$.
- When the functions $(B_\ell^L)_{1 \le \ell \le L}$ form an orthonormal family, $\|\hat{d}^L\|_\infty$ increases with $L$ until $L = L^*$ and then $\|\hat{d}^L\|_\infty$ is constant with $L$.

Their behavior will be illustrated with different function families in Section 6.

## 3.2 Asymptotic confidence band for $f^L$

Note that in the asymptotic framework $n \to \infty$, the previous definition of $\hat{d}^L$ induces a natural asymptotic confidence band for the function $f^L$. Indeed, we can prove that

**Theorem 3.2.** *Set Assumption 1 and Assumption 3 and a probability $\alpha \in [0, 1]$. Then, we have,*

$$\lim_{n \to +\infty} \mathbb{P}\left(\forall t \in [0, 1], |\underline{\hat{f}}^L(t) - f^L(t)| \leq \hat{d}^L(t)\right) \geq 1 - \alpha,$$

with $\hat{d}^L(t) = \hat{q}^L \sqrt{\hat{C}^L(t,t)/N}$ and $\hat{q}^L$ is defined as the solution of Equation 4.

The proof is given in Appendix.

Then a confidence band for $f^L$ at the asymptotic confidence level $1 - \alpha$ for a large number of observations $n$ is given by

$$CB(f^L) = \left\{\forall t \in [0, 1], \left[\underline{\hat{f}}^L(t) - \hat{d}^L(t); \underline{\hat{f}}^L(t) + \hat{d}^L(t)\right]\right\}.$$

# 4  Method 1: Confidence Band of $f^{L^*}$ by correcting the bias

The function of interest is $f^{L^*} = \underline{f}^{*,L^*}$, rather than $\underline{f}^L$. Therefore, our objective is to construct a confidence band for $f^{L^*}$. However, an unbiased estimator of $f^{L^*}$ is not available by definition, since the true dimension $L^*$ is unknown. We propose instead to work with the estimator $\underline{\hat{f}}^L$ and to debias the corresponding confidence band.

To do this, we use the decomposition between the bias term and the statistical term, outlined in Equation 2. The idea is to bound the infinite norm of these two terms. A first strategy is to bound each term separately, then add the two bounds to construct the confidence band. However, this approach tends to produce a band that is too large and too conservative. This is because applying the infinite norm to each term before bounding them does not take into account the functional nature of the two terms.

A second strategy is to retain the functional aspect by bounding the infinity norm of the sum of the two functional terms. This approach is detailed in this section.

In Section 4.1, we first rewrite the band as a band around $\underline{\hat{f}}^L(t)$. We need to estimate the band bound and the bias. To do this, we divide the sample into two sub-samples. This choice is not ideal because it increases the variability of the estimators. But at least, it provides independence between the estimators of the two quantities, which makes it possible to establish the theoretical coverage of the final band. More precisely, we use a first subsample $\mathbf{y}^1$ to estimate the bound defined in Section 3. A second subsample $\mathbf{y}^2$ is used to estimate the bias term (without the infinite norm). This results in a pointwise correction of the bias, and the final confidence band is centered around $\underline{\hat{f}}^{L_{\max}, L^*}$.

This procedure provides a collection of confidence bands, for $L \in \{L_{\min}, \dots, L_{\max}\}$ with variable width. Then, in Section 4.2, we propose a criterion to select the "best" band by minimizing its width. We discuss the band thus obtained at the end of the section and its limits.

## 4.1 Construction of the band of $f^{L^*}$ for a given $L$

We introduce two independent sub-samples $\mathbf{y}^1$ and $\mathbf{y}^2$ of $\mathbf{y}$ of length $N_1$ and $N_2$ such that $N_1 + N_2 = N$.

354 We use the first sub-sample $\mathbf{y}^1$ to calculate $\hat{\underline{f}}_1^L(t)$, an estimator of $\underline{f}^L(t)$ and a functional bound

355 denoted $\hat{d}_1^L$ that controls the bias term $\underline{f}^L(t) - \hat{\underline{f}}_1^L(t)$. This bound is defined in Section 3 applied on

356 $\mathbf{y}^1$, for a given level $\alpha$, such that:

$$\mathbb{P}\left(\forall t \in [0,1], -\hat{d}_1^L(t) \le \underline{f}^L(t) - \hat{\underline{f}}_1^L(t) \le \hat{d}_1^L(t)\right) = 1 - \alpha. \tag{5}$$

357 Next, we need to control the bias $Bias_L(t) = \underline{f}^L(t) - f^{L^*}(t)$. Recall that when $L_{\max}$ is sufficiently large

358 and $n > L_{\max}$, we have $f^{L^*} = \underline{f}^{L_{\max}, L^*}$. We therefore need to control the $Bias_L(t) = \underline{f}^L(t) - \underline{f}^{L_{\max}, L^*}(t)$.

359 It would be tempting to replace $Bias_L(t)$ by its estimation based on the second sample $\mathbf{y}^2$, but this

360 would introduce an estimation error that we also need to control, in the same spirit as what is

361 done in Lacour, Massart, and Rivoirard (2017). We can again use Section 3 to compute the function

362 $\hat{d}_2^{L, L_{\max}}(t)$ on the sample $\mathbf{y}^2$, and the functional estimators $\hat{\underline{f}}_2^L(t)$ and $\hat{\underline{f}}_2^{L_{\max}, L^*}(t)$ of $\underline{f}^L(t)$ and $\underline{f}^{L_{\max}, L^*}(t)$,

363 respectively. This allows us to construct the following band for $\underline{f}^L(t) - \underline{f}^{L_{\max}, L^*}$ for a confidence level

364 $1 - \beta$,

$$\mathbb{P}\left(\forall t \in [0,1], -\hat{d}_2^{L, L_{\max}}(t) \le \underline{f}^{L_{\max}, L^*}(t) - \underline{f}^L(t) - (\hat{\underline{f}}_2^{L_{\max}, L^*}(t) - \hat{\underline{f}}_2^L(t)) \le \hat{d}_2^{L, L_{\max}}(t)\right) = 1 - \beta. \tag{6}$$

365 Combining Equation 5 and Equation 6, we can provide a debiased confidence band of $f^{L^*}(t)$.

366 **Proposition 4.1.** *Let us define*

$$\hat{\theta}_1^L(t) := -\hat{d}_1^L(t) - \hat{d}_2^{L, L_{\max}}(t) + \hat{\underline{f}}_2^{L_{\max}, L^*}(t) - \hat{\underline{f}}_2^L(t)$$
$$\hat{\theta}_2^L(t) := \hat{d}_1^L(t) + \hat{d}_2^{L, L_{\max}}(t) + \hat{\underline{f}}_2^{L_{\max}, L^*}(t) - \hat{\underline{f}}_2^L(t),$$

367 *where $\hat{d}_1^L(t)$ is defined on sample $\mathbf{y}^1$ by Equation 5 for a level $\alpha$ and $\hat{d}_2^{L, L_{\max}}(t)$ is defined on sample $\mathbf{y}^2$ by*

368 *Equation 6 for a level $\beta$. Then we have*

$$\mathbb{P}\left(\forall t \in [0,1], \quad \hat{\theta}_1^L(t) \le f^{L^*}(t) - \hat{\underline{f}}_1^L(t) \le \hat{\theta}_2^L(t)\right) \ge 1 - \alpha\beta.$$

369 The proof is given in Appendix.

370 This defines a confidence band which can be defined either around $\hat{\underline{f}}_1^L$:

$$CB_2(\underline{f}^{L^*}) = \left\{\forall t \in [0,1], \left[\hat{\underline{f}}_1^{L, L^*}(t) + \hat{\theta}_1^L(t) ; \hat{\underline{f}}_1^{L, L^*}(t) + \hat{\theta}_2^L(t)\right]\right\}$$

371 or around $\hat{\underline{f}}_2^{L_{\max}, L^*}$:

$$CB_2(\underline{f}^{L^*}) = \left\{\forall t \in [0,1], \left[\hat{\underline{f}}_2^{L_{\max}, L^*}(t) + \bar{\theta}_1^L(t) ; \hat{\underline{f}}_2^{L_{\max}, L^*}(t) + \bar{\theta}_2^L(t)\right]\right\}.$$

372 with $\bar{\theta}_1^L(t) = \hat{\underline{f}}_1^L(t) - \hat{\underline{f}}_2^{L, L^*}(t) - \hat{d}_1^L(t) - \hat{d}_2^{L, L_{\max}}(t)$ and $\bar{\theta}_2^L(t) = \hat{\underline{f}}_1^L(t) - \hat{\underline{f}}_2^{L, L^*}(t) + \hat{d}_1^L(t) + \hat{d}_2^{L, L_{\max}}(t)$.

373 *Remark* 4.1. The two functions $\hat{d}_1^L(t)$ and $\hat{d}_2^{L, L_{\max}}(t)$ are of the same order because they are constructed

374 using the same approach. They depend on the length of the samples. To obtain the thinnest band,

375 the best strategy is to divide the sample in two sub-samples of equal length $N_1 = N_2 = N/2$.

376 The behavior of $\hat{d}_1^L$ was described in Section 3. Let us describe the behavior of $\hat{d}_2^{L,L_{\max}}$:

377 • $\|\hat{d}_2^{L,L_{\max}}\|_\infty$ decreases with $L$.

378 • When $L > L^\varepsilon$, $\|\hat{d}_2^{L,L_{\max}}\|_\infty$ is constant with $L$ and the probability in Equation 6 is equal to 1.

379 • When $L^\star < L < L^\varepsilon$, $\|\hat{d}_2^{L,L_{\max}}\|_\infty$ is constant with $L$ when the functions $B_\ell^L$ form an orthonormal
380 family. Otherwise, the behavior is erratic.

381 This means that when the band defined in Proposition 4.1 is calculated for $L > L^\varepsilon$, the confidence
382 level is $1 - \alpha$ instead of $1 - \alpha\beta$.

383 The advantage of this approach is that the band bias is corrected and the level for the true function $f^{L^\star}$
384 is guaranteed when $L^\varepsilon$ is large. This was the main aim of the paper. The main limit of this approach
385 is that the band is constructed with samples with half sizes, leading to less precision. This will be
386 illustrated in Section 6. Nevertheless, this method gives finer confidence bands than cross-validation,
387 and with the right level of confidence.

388 A natural question is then the choice of the dimension $L$. This is the purpose of the next section.

### 4.2 Influence of $L$

390 This approach produces a collection of debiased confidence bands for different values of $L$. The
391 confidence bands have different widths but the same confidence level $1 - \alpha\beta$. It is therefore natural to
392 want to select one of them. This means we want to select the best dimension $L$ among the collection
393 $\{L_{\min}, \ldots, L_{\max}\}$. We need to define what "best" means. It is quite intuitive to want to focus on the
394 finest band, fine in the sense of a certain norm. Here we consider the infinite norm of the width
395 of the confidence band. This gives preference to smooth bands. We therefore define the following
396 criteria for selecting $L$.

$$\hat{L} = \arg\min_L \left\{ \sup_t |\hat{\theta}_2^L(t) - \hat{\theta}_1^L(t)| \right\} = \arg\min_L \left\{ \sup_t |\hat{d}^L(t) + \hat{d}^{L,L_{\max}}(t)| \right\}. \tag{7}$$

397 This global approach guarantees that each band of the collection is debiased and then the dimension
398 is selected. It will be illustrated in Section 6.

399 In the next section, instead of debiasing each band, we employ another strategy focusing on the
400 construction of a selection criteria that will guarantee that the bias is negligible.

## 5 Method 2: Selection of the best confidence band with a criteria taking into account the bias

403 In this section, we propose a new method in the non asymptotic setting to provide a confidence band
404 of $f^{L^\star}$ without correction the bias but taking it into account in the selection procedure.

405 Our method uses the collection of confidence bands defined in Section 3. Instead of correcting their
406 bias, the strategy is to propose a selection criterion that is a trade-off between this bias and the
407 dimension of the basis. To do this, we propose a new heuristic criterion linked to the definition of
408 the band itself, considering the estimation of the band as the estimation of a quantile of a certain
409 empirical process. The criterion is inspired by model selection tools for choosing the best dimension
410 $L$. In the following, we assume that $L_{\max}$ is large enough such that $\underline{f}^{L_{\max},L^\star} = f^{L^\star}$.

411 We work on the quantile $q^L$ introduced in Equation 3, its oracle version $q^{L^\star}$ for level $L^\star$ and its
412 estimate $\hat{q}^L$. All are scalars, belonging to a collection indexed by $L \in \{L_{\min}, \ldots, L_{\max}\}$. A natural

criterion for choosing the best $L$ is that the estimator $\hat{q}^L$ minimizes the quadratic error $\mathbb{E}\left(\|q^{L^*} - \hat{q}^L\|^2\right)$. However, this quadratic error is unknown as $q^{L^*}$ is unknown. We cannot use it directly.

Instead, we study $\|\hat{q}^{L_{\max}} - \hat{q}^L\|^2$. While the theoretical quadratic error $\mathbb{E}\left(\|q^{L^*} - \hat{q}^L\|^2\right)$ decreases when $L < L^*$ and increases when $L > L^*$, the $\|\hat{q}^{L_{\max}} - \hat{q}^L\|^2$ approximation to this error always decreases when $L > L^*$, as illustrated in Section 6.

We see a behavior similar to a bias, high when the dimension is small, and small when the dimension is large. Selecting a dimension using this criterion will always overfit the data. We therefore propose to penalize this quantity by the dimension $L$ divided by the sample size $N$, as usual in model selection criteria. To do this, we introduce a regularisation parameter $\lambda > 0$ which balances the two terms. A natural criterion to select the best $L$ is then

$$\widetilde{crit}(L) = \|\hat{q}^{L_{\max}} - \hat{q}^L\|^2 + \lambda \frac{L}{N}.$$

Then we define

$$\tilde{L} = \arg\min_L \widetilde{crit}(L),$$

and take the band centered around $\underline{f}^{\tilde{L},L^*}$:

$$CB_3(\underline{f}^{L^*}) = CB_1(\underline{f}^{\tilde{L},L^*})$$

This criterion is illustrated in Section 6.5. We also compare with two other standard approaches, heuristic as well, namely the cross-validation approach used to select the dimension $L$ which minimizes the reconstruction error, and a thresholding method which keeps the higher dimension $L$ with large enough coefficients. These two methods are less oriented to the objective of controling the level of the selected confidence band.

# 6   Simulation study

In this section, we illustrate the different statements provided along the paper on generated data. First, in Section 6.1, we describe the generating data process and illustrate the linear estimator considered in this paper. In Section 6.2, we illustrate the first confidence band, for a fixed level, as introduced in Section 3. Then, we illustrate the debiased confidence band in Section 6.3, and discuss the model selection criterion in Section 6.4, comparing both of them with state-of-the-art methods in Section 6.5. We finally study the generalization of the method out of the class of models in Section 6.6.

## 6.1   Generating data process

To illustrate the model, we simulate a regression functional model with $n = 40$ regular timepoints per individual and $N = 25$ individuals. In Figure 1, the function $f$ (black curve) belongs to the Fourier (resp. Legendre and Spline) family with $L^* = 7$ and the noisy individual observations $(y_{ij})_{1 \leq i \leq N, 1 \leq j \leq n}$ (grey curves) have a functional noise in dimension $L^\varepsilon = 15$, also in the Fourier (resp. Legendre and Spline) family on the left plot (resp. middle and right).

## 6.2   Confidence band for a fixed level

The general band for $f^L$ derived in Theorem 3.1 is illustrated on Figure 2. It displays on the top row several functional data generated under either the Fourier family (left), Legendre (middle) or Spline (right), on the middle row the confidence bands of $\underline{f}^L$ for different values of $L \in \{3, 5, 7, 11\}$ and 15,

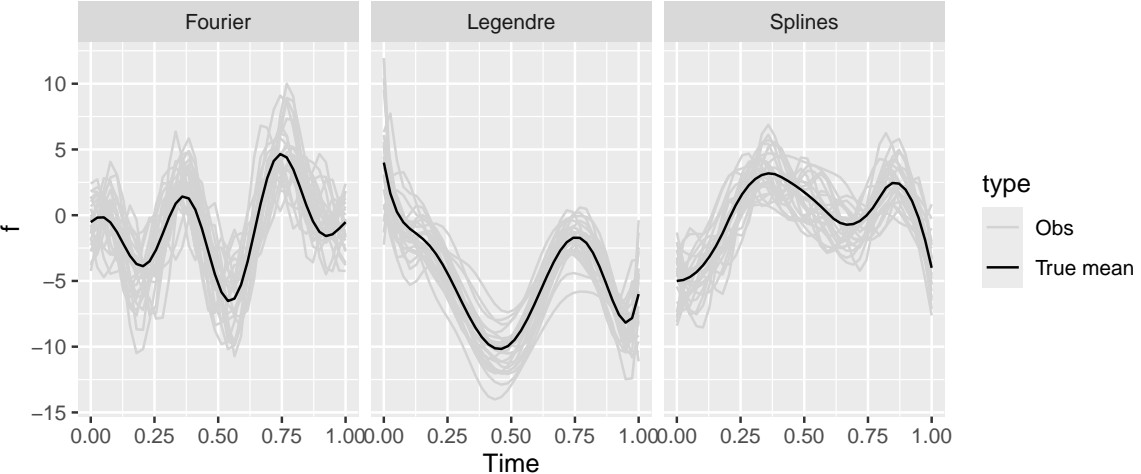

Figure 1: Functional regression with different bases. We generate a regression functional model using three different basis families: Fourier (left), Legendre (middle), and Splines (right). The black curve represents the true mean function, while the gray curves show individual noisy observations.

and on the bottom row the bound $\hat{d}^L$. On the middle row, the true functions $\underline{f}^L$ are displayed in black and the confidence bands in color. The bands are very precise for each $L$. The behavior of $\hat{d}^L$ increases with $L$. As $d^L$ can be seen as a variance, $\hat{d}^L(t)$ is larger on the boundary of the time domain, as there are less observations near 0 and 1.

We also evaluate numerically the levels of the obtained confidence bands. For this, 1000 datasets are simulated, the confidence band is estimated for each of them. The empirical confidence level is then evaluated as the proportion of confidence bands that contain the true function over 5000 test timepoints. Table 1 presents the empirical confidence levels for different values of $L$ and several sample sizes $n \in \{40, 150\}$, and $N \in \{10, 25, 60\}$, and for the 3 basis. When $N = 25$ and $N = 60$, the level is the expected one whatever the value of $L$. We will see in the next sections that this will not be the case for the debiased confidence band. When $N = 10$, the level is too small, especially when $L$ is large. This might be due to the the large number of parameters to be estimated in the covariance matrix, with a small number of observations $N$.

## 6.3   Method 1: confidence bands by correcting the bias

We illustrate the confidence band of $f^{L^*}$ given in Proposition 4.1. In Figure 3, top row, we plot the confidence bands obtained for different dimensions $L \in \{3, 5, 7, 11, 15\}$ with Fourier, Legendre and Splines families and $\alpha = \beta = \sqrt{0.05} \approx 0.22$. We can see that all the confidence bands are alike. Especially, they are unbiased, even for $L = 3$. A larger dimension $L$ provides a smoother band. On the middle and bottom rows of Figure 3, we illustrate the two terms that enter the confidence band, $t \mapsto \hat{d}_1^L(t)$ and $t \mapsto \hat{d}_2^{L,L_{\max}}(t)$. Their behavior is the same along time. The function $\hat{d}_1^L(t)$ can be seen as a variance, this is why it is larger near 0 and 1 where there are less observations. The function $\hat{d}_2^{L,L_{\max}}(t)$ is smaller than $\hat{d}_1^L(t)$ because it controls the remaining rest after the projection. Note that as expected when $L > L^\varepsilon$, $\hat{d}_2^{L,L_{\max}}(t)$ is close to 0. As explained before, the influence of $L$ is not the same for the two functions. When $L$ increases, $\hat{d}_1^L(t)$ increases while $\hat{d}_2^{L,L_{\max}}(t)$ decreases.

In Table 2, we simulate 1000 repeated datasets with the Legendre family and with two sample sizes $n = 40$ and $n = 150$ and $N = 25$. For each dataset, we compute the confidence band defined in Proposition 4.1 with a theoretical confidence level of $1 - \alpha\beta = 0.95$ and for different values of $L$.

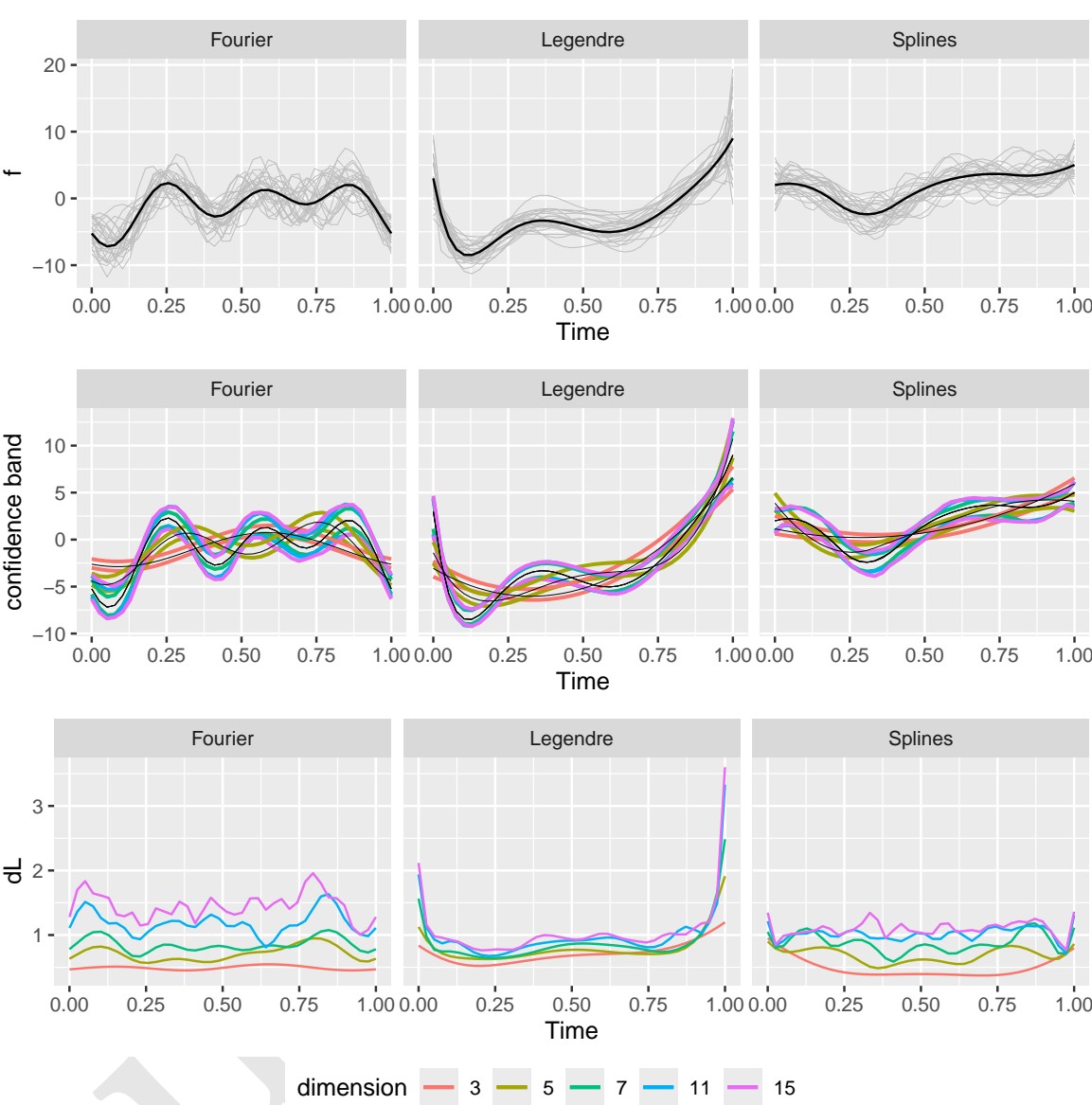

Figure 2: Confidence bands for $f^L$. For the three basis families (Fourier, on the left, Legendre, on the middle, and Splines on the right) we display: (top row) the observed functional data; (middle row) the estimated confidence bands for increasing values of L (3, 5, 7, 11 and 15); and (bottom row) the associated bound dL.

Table 1: Empirical coverage of CB for different basis families. The empirical confidence level of the confidence band CB is estimated over 1000 repetitions, for alpha = 0.05. Confidence bands are calculated with the Fourier (top), Legendre (middle) and Splines (bottom) basis families. Each row corresponds to a different number of basis functions L, and each column to a different pair of sample sizes (n,N).

| L | n/N | | | | | |
|---|---|---|---|---|---|---|
| | 40/10 | 150/10 | 40/25 | 150/25 | 40/60 | 150/60 |
| 3 | 0.930 | 0.933 | 0.939 | 0.942 | 0.952 | 0.951 |
| 5 | 0.921 | 0.913 | 0.927 | 0.927 | 0.950 | 0.951 |
| 7 | 0.926 | 0.926 | 0.931 | 0.930 | 0.948 | 0.950 |
| 11 | 0.921 | 0.911 | 0.933 | 0.933 | 0.951 | 0.951 |
| 15 | 0.915 | 0.901 | 0.926 | 0.927 | 0.943 | 0.943 |

| L | n/N | | | | | |
|---|---|---|---|---|---|---|
| | 40/10 | 150/10 | 40/25 | 150/25 | 40/60 | 150/60 |
| 3 | 0.926 | 0.928 | 0.935 | 0.937 | 0.947 | 0.945 |
| 5 | 0.925 | 0.926 | 0.928 | 0.937 | 0.946 | 0.943 |
| 7 | 0.924 | 0.929 | 0.927 | 0.937 | 0.923 | 0.939 |
| 11 | 0.927 | 0.924 | 0.923 | 0.931 | 0.936 | 0.942 |
| 15 | 0.929 | 0.929 | 0.931 | 0.935 | 0.942 | 0.944 |

| L | n/N | | | | | |
|---|---|---|---|---|---|---|
| | 40/10 | 150/10 | 40/25 | 150/25 | 40/60 | 150/60 |
| 3 | 0.929 | 0.930 | 0.927 | 0.931 | 0.936 | 0.939 |
| 5 | 0.921 | 0.917 | 0.943 | 0.944 | 0.941 | 0.946 |
| 7 | 0.911 | 0.915 | 0.939 | 0.942 | 0.952 | 0.950 |
| 11 | 0.913 | 0.911 | 0.948 | 0.951 | 0.952 | 0.949 |
| 15 | 0.903 | 0.900 | 0.950 | 0.946 | 0.952 | 0.955 |

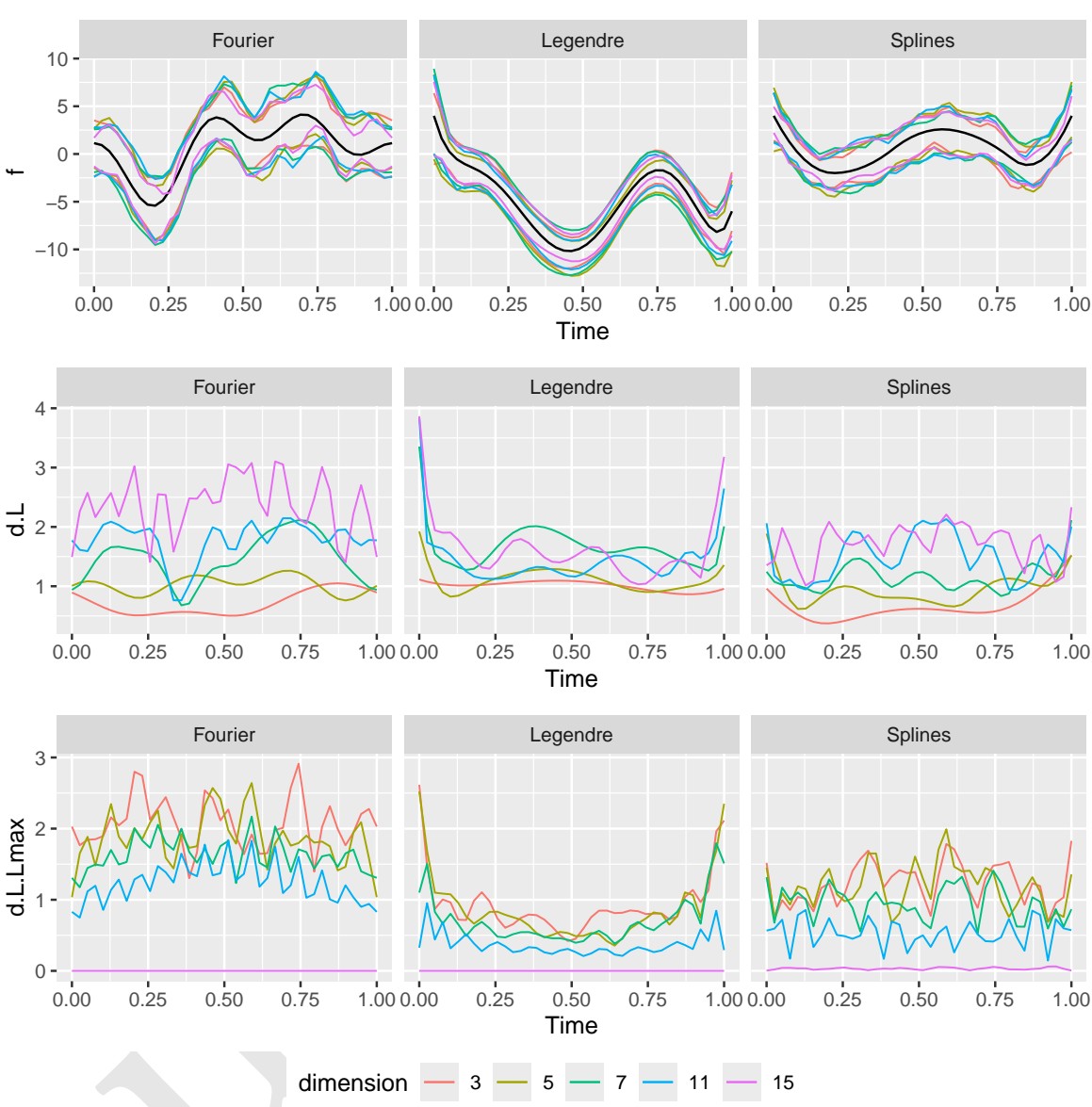

Figure 3: Visualization of confidence bands by correcting the bias. For a given dataset, we plot several confidence bands (top row), functions dL (middle row) and dLLmax (bottom row). Bands and functions are estimated with Fourier (left column), Legendre (middle column) and Spline (right column) basis and several dimensions L (3, 5, 7, 11, 15), and Lmax = 25.

Table 2: Empirical coverage of CB2. The table reports the empirical level of confidence of the proposed confidence band CB2, computed for various values of L (rows) and n (columns), with fixed N=25, and various basis (top: Fourier, middle: Legendre, bttom: Splines). The nominal confidence level is set to 0.05, using parameters alpha=beta=0.2, such that the combined coverage is 1-0.95.

| L | n | |
|---|---|---|
| | 40 | 150 |
| 3 | 0.958 | 0.958 |
| 5 | 0.966 | 0.966 |
| 7 | 0.980 | 0.980 |
| 11 | 0.983 | 0.983 |
| 15 | 0.723 | 0.723 |

| L | n | |
|---|---|---|
| | 40 | 150 |
| 3 | 0.925 | 0.925 |
| 5 | 0.948 | 0.948 |
| 7 | 0.951 | 0.951 |
| 11 | 0.923 | 0.923 |
| 15 | 0.747 | 0.747 |

| L | n | |
|---|---|---|
| | 40 | 150 |
| 3 | 0.953 | 0.953 |
| 5 | 0.973 | 0.973 |
| 7 | 0.963 | 0.963 |
| 11 | 0.910 | 0.910 |
| 15 | 0.728 | 0.728 |

Then the confidence level is approximated as the proportion of confidence bands containing the true function $f$ over 5000 test timepoints. Remark that when $L < L^\varepsilon$, the level is the expected one, that is 0.95. When $L > L^\varepsilon$, the level is no more ensured, as explained before. Indeed the term $d^{L,L^{\max}}$ is mainly equal to 0 when $L^{\max}$ is large enough, and the level is close to $1 - \alpha$ instead of $1 - \alpha\beta$. This is not the case for the band in Section 3, as this is due to the correction of the bias.

We illustrate the different terms involved in Equation 7 in Figure 4: we plot for a given dataset, the infinity norm of the width of the band $\hat{d}^L(t) + \hat{d}^{L,L_{\max}}(t)$ (top), of $\hat{d}^L(t)$ (middle) and $\hat{d}^{L,L_{\max}}(t)$ (bottom) functions obtained with the Fourier (left column), Legendre (middle column) and Spline (right column) basis. As already said, $\|\hat{d}^L\|_\infty$ increases with $L$ while $\|\hat{d}^{L,L_{\max}}\|_\infty$ decreases (and is zero when $L > L^\varepsilon$). The width of the band wrt $L$ does not have a $U$-shape, as expected. It is thus difficult to minimize this criterion and the selection of $\hat{L}$ is thus not stable. But again, whatever the value of $\hat{L}$, the corresponding band is debiased in the collection. We will see in the next sections that its width is smaller than standard approaches. The performance of the selection is also illustrated in the next section.

## 6.4 Method 2: a model selection criterion to take into account the bias

In Figure 5, we illustrate the behavior of the selection criterion introduced in Section 5 on simulated data, with $\lambda \in \{0, 0.5, 1, 2\}$ for the three basis. We can see that $\tilde{L}$ is overestimated. When considering

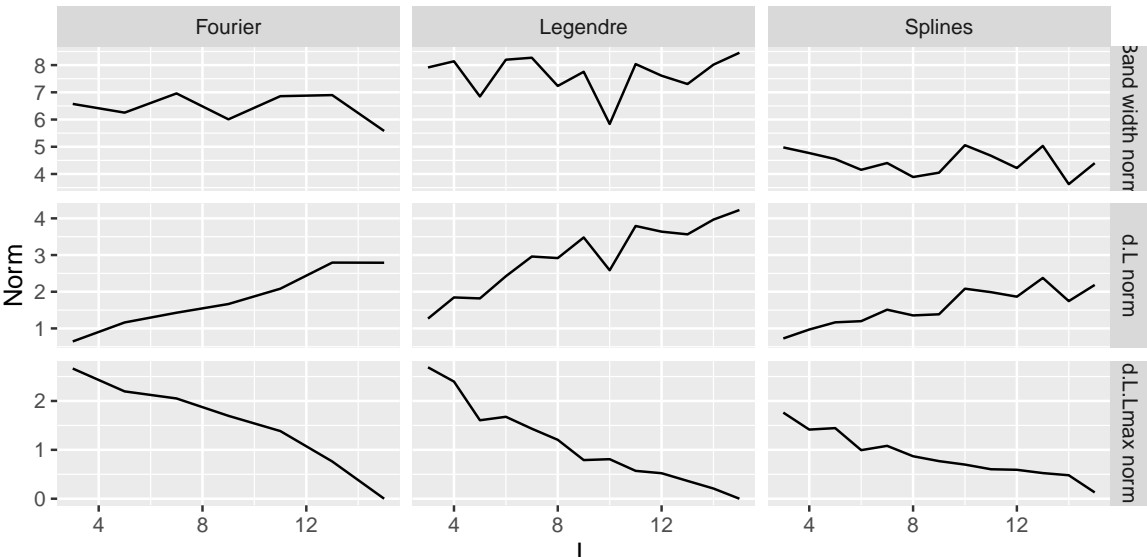

Figure 4: Norm of the confidence band width and associated bounds. For a given dataset, we compute: (top row) the norm of the width of the confidence band; (middle row) the norm of the function dL; and (bottom row) the norm of the function dLLmax, for various dimensions L. Results are shown for the Fourier (left column), Legendre (middle column), and Splines (right column) basis families.

nested spaces, it ensures that $\tilde{L}$ tends to be larger than $L^*$ and thus the confidence band is automatically unbiased.

## 6.5 Comparison of the methods with the state-of-the-art

We evaluate the performance of the two selection criteria introduced in this paper, and compare the two strategies $CB_2$ and $CB_3$ with some standard approaches. More precisely, we simulate 1000 repeated datasets. The different confidence bands and the norm of their widths are computed for several $L$. We apply the selection criteria and plot the distribution of the estimated dimension in Figure 6, for the three basis families, for several model selection criteria: $\hat{L}$, $\tilde{L}$, cross validation and hard thresholding. The dimension $\hat{L}$ is almost always larger than the true $L^* = 7$. The fact that it is larger is not a problem because the selected band is unbiased and has the correct level as soon as $L^\varepsilon$ is large. However, the criterion tends to select a band that is (too) smooth. We can see that the selected dimension $\tilde{L}$ is smaller in distribution, and closer to the true value than $\hat{L}$. In addition, as we then use the confidence band of Section 3, the confidence level is guaranteed to be as expected. The model selected by cross validation is rather good, for all the basis considered. On the other hand, the model selected by hard thresholding is not good, particularly for a non orthonormal basis.

The reformulation of the band around $\underline{f}_2^{L_{\max},L^*}$ is close to the band presented in Section 3 for $L = L_{\max}$, that is a band centered around $\underline{f}^{L_{\max},L^*}$. A natural question is to understand what is the gain by doing so instead of using the band from Section 3 with $L = L_{\max}$, namely the band $\left[\underline{f}^{L_{\max},L^*}(t) - \hat{d}^{L_{\max}}(t); \underline{f}^{L_{\max},L^*}(t) + \hat{d}^{L_{\max}}(t)\right]$. To do that, we have to understand the behavior of the different terms. As it is difficult to compare theoretically the width of the two bands, we compare them using simulations. For 1000 repeated datasets, we compute several confidence bands: the $CB_1$ band constructed in Section 3 with $L_{\max}$, the $CB_2$ band defined in Proposition 4.1 with $\hat{L}$, the $CB_3$ band defined in Section 5 with $\tilde{L}$ and the ideal (and not accessible) band constructed in Section 3 with the true $L^*$. In Figure 7, we present boxplots of the norms of the band width with $\hat{L}$, $L_{\max}$, $L^*$ and

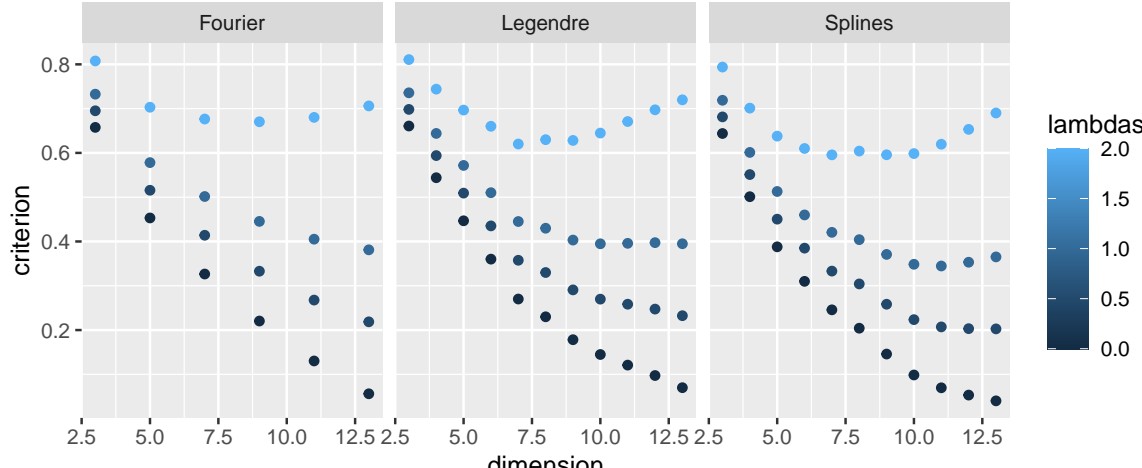

Figure 5: Behavior of selection criteria as a function of model dimension. For a given simulated dataset, we display the evolution of the selection criteria with respect to the dimension L, for several values of the regularization parameter lambda. Results are shown for the Fourier (left), Legendre (middle), and Splines (right) basis families.

Table 3: Comparison with the state-of-the-art for the empirical confidence level. The table reports the performance of the proposed confidence band CB3, estimated over 1000 repetitions, for various model selection criteria and competitive methods (rows) and for various basis families (columns). Results are shown for alpha=0.05, with n=40 and N=25.

| model | Fourier | Legendre | Splines |
|---|---|---|---|
| Lstar | 0.931 | 0.932 | 0.940 |
| Lmax | 0.926 | 0.932 | 0.949 |
| Lhat | 0.972 | 0.924 | 0.904 |
| Ltilde | 0.933 | 0.926 | 0.926 |
| L.L0 | 0.895 | 0.922 | 0.901 |
| L.CV | 0.913 | 0.918 | 0.831 |
| Mean | 0.926 | 0.932 | 0.950 |
| FFSCB | 0.922 | 0.941 | 0.950 |

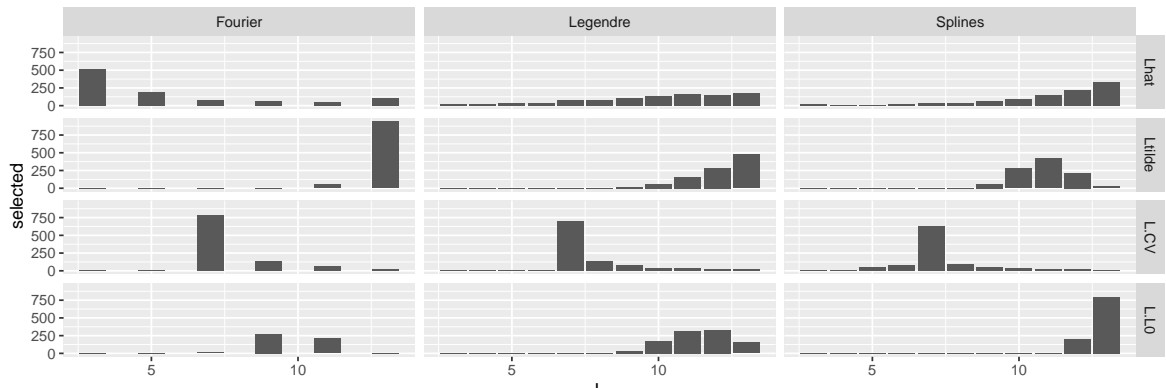

Figure 6: Distribution of the selected model dimension across selection methods. Based on 1000 simulated datasets, we report the distribution of the estimated dimension L for four model selection methods: from top to bottom — the debiased confidence band CB2 with Lhat, the band CB3 with Ltilde , cross-validation, and hard-thresholding. The true model dimension is Lstar = 7.

$\tilde{L}$. We also use the empirical mean and the method FFSCB Liebl and Reimherr (2023). The width of the confidence band with the true $L^*$ is smaller, which is expected but unfortunately not achievable. The width of the confidence band $CB_1$ is smaller than that of the band $CB_2$. This can be explained by the fact that we estimate two different quantities, on smaller datasets, for more conservative levels $(1 - \alpha$ and $1 - \beta$ respectively) in order to finally achieve the confidence level of $1 - \alpha\beta$. This also explains why the cross validation and hard-thresholding methods, which also divide the sample into two parts, do not give good results either. The model given by the heuristic model selection criterion $\widetilde{crit}$ achieves good performance. Note that the width of the selected model $\tilde{L}$ is better than the width of the confidence band with a large level $L_{\max}$, which one should have used to avoid model selection. The empirical mean and the band given by FFSCB are a bit larger.

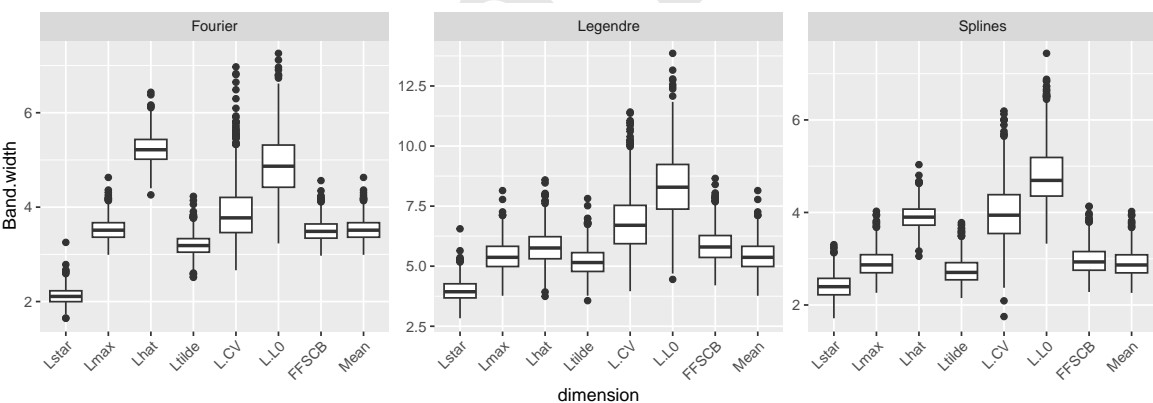

Figure 7: Width of the confidence bands accross selection methods. Boxplots show the distribution of the confidence band width over 1000 repetitions, for the dimension selected by the various criteria introduced in this paper (as well as standard baselines). Results are shown for the Fourier (left), Legendre (middle), and Splines (right) basis families.

## 6.6 Generalization out of the model

We now illustrate the behaviour of the bands when the basis used for estimation is poorly specified. We simulate 1000 data sets with a spline basis and estimate the confidence bands with the Fourier and Legendre basis, for different values of $n$ and $N$. The coverage rates are presented in Table 4.

Table 4: Empirical coverage of confidence bands under model misspecification. The empirical confidence level is estimated over 1000 repetitions. Data are generated using a Splines basis, while confidence bands are computed using the Fourier (top) and Legendre (bottom) basis families. Each row corresponds to a different value of L, and each column to a different pair of sample sizes (n,N).

| L | n/N | | | |
|---|---|---|---|---|
| | 50/10 | 150/10 | 50/40 | 150/40 |
| 5 | 0.057 | 0.057 | 0.016 | 0.017 |
| 7 | 0.113 | 0.107 | 0.042 | 0.039 |
| 11 | 0.232 | 0.205 | 0.096 | 0.077 |
| 15 | 0.298 | 0.254 | 0.153 | 0.118 |
| L | n/N | | | |
| | 50/10 | 150/10 | 50/40 | 150/40 |
| 5 | 0.133 | 0.142 | 0.031 | 0.035 |
| 7 | 0.684 | 0.679 | 0.415 | 0.429 |
| 11 | 0.914 | 0.910 | 0.939 | 0.940 |
| 15 | 0.916 | 0.912 | 0.951 | 0.951 |

Table 5: Empirical coverage of confidence bands across selection methods under model misspecification. The empirical confidence level is estimated over 1000 repetitions. Data are generated using a Splines basis, while confidence bands are computed using the Fourier (top) and Legendre (bottom) basis families. The rows correspond to different model selection criteria and dimensions L, and the columns to various combinations of sample sizes (n,N).

| L | n=40, N=25 | n=150, N=25 |
|---|---|---|
| L tilde | 0.098 | 0.117 |
| L.CV | 0.090 | 0.084 |
| L | n=40, N=25 | n=150, N=25 |
| L tilde | 0.942 | 0.950 |
| L.CV | 0.588 | 0.593 |

The Fourier basis does not give a correct rate. On the other hand, the Legendre basis gives very satisfactory coverage rates for $L > 11$.

Next, we illustrate the $\tilde{L}$ dimension selection method and compare it to the cross-validation method. Table 5 presents the coverage rates of the corresponding confidence bands estimated with the Fourier and Legendre basis, in the case $N = 25$ and $n \in \{40, 150\}$. Once again, we see that the Fourier basis does not give good results, either by cross-validation or by $\tilde{L}$. On the other hand, with the Legendre basis, the $\tilde{L}$ method gives a satisfactory coverage rate, even if it is underestimated, whereas the cross-validation method is very poor. Moreover, the widths of the confidence bands selected with $\tilde{L}$ and by cross validation are represented by boxplot in Figure 8. It can be seen that the cross-validation approach provides wider bands, even though their confidence level is not guaranteed. The method proposed in this paper provides a narrower band with a correct level of confidence. We thus recommend to use the Legendre family with the criteria $\tilde{L}$.

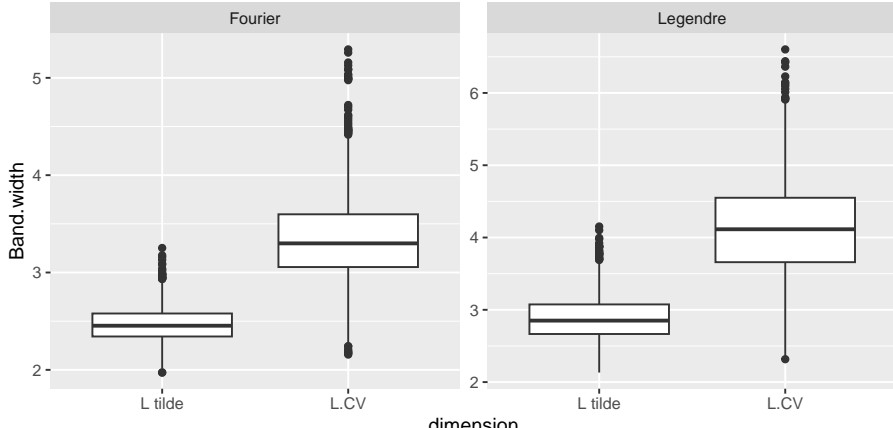

Figure 8: Width of confidence bands with model selection under model misspecification. The average width of the confidence bands is evaluated over 1000 repetitions. Data are generated using a Splines basis, while bands are computed with the Fourier (top) and Legendre (bottom) basis families. Rows correspond to combinations of model selection criteria and dimensions L, and columns to different sample size pairs (n,N).

## 7 Real data analysis

In this section, we illustrate the proposed method on the Berkeley Growth Study data. It consists of the heights in centimeters of 39 boys at 31 ages from 1 to 18. We approximate these curves by the 3 basis Legendre, Splines and Fourier. We select the level of each basis using the method introduced in Section 5.

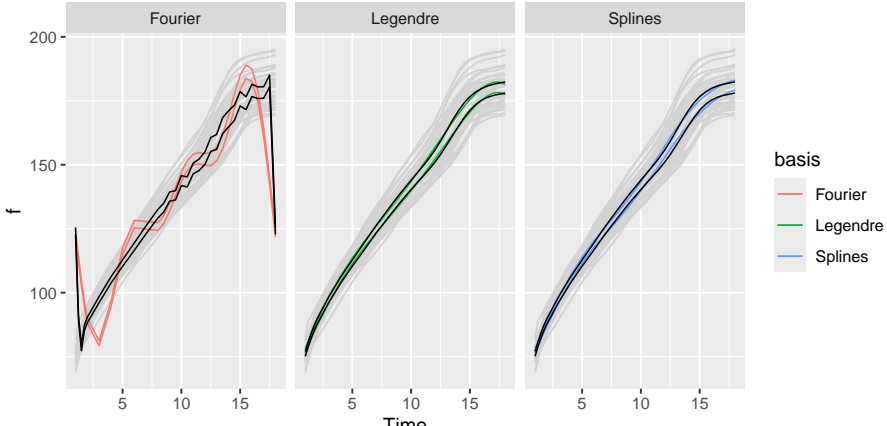

Figure 9: Real data analysis example. We display the confidence bands for Fourier (left), Legendre (middle) and Splines (right) basis on the Berkeley Growth Study data. Black curves correspond to the confidence bands with $L_{max}$, while colored one are the confidence bands with L tilde.

In Figure 9, we display the confidence bands associated with Section 3 in black and those associated with Section 5, for the three basis. As the data is not periodic, the Fourier basis is meaningless, as is the associated confidence band, whatever the dimension considered. Splines and Legendre basis give similar confidence bands. Analyzing the width of the bands in Table 6, compared to that obtained with $L_{max}$, we find that they are less smooth but also smaller, and from our empirical study we guess that it makes a trade-off between bias and variance.

Table 6: Real data analysis example, Berkeley Growth Study data. We display the width of the confidence bands for Fourier, Legendre and Splines basis for the confidence band of Section 3 with Lmax and the confidence band of Section 5. We also precise the dimension of the selected model.

|  | Basis | | |
| --- | --- | --- | --- |
|  | Legendre | Splines | Fourier |
| Width Lmax | 2.12 | 2.12 | 2.20 |
| Width selected | 1.99 | 1.95 | 2.08 |
| Model selected | 6.00 | 5.00 | 7.00 |

## 8 Conclusion

This paper discusses the construction of confidence bands when considering a functional model. Depending on the nature of the family (an orthogonal or orthonormal basis, or simply a vector space), the theoretical guarantees of the linear estimator are recalled and illustrated. Several confidence bands are then proposed. An extensive experimental study on Fourier, Legendre, and Spline basis illustrates the theoretical and methodological propositions, and a real data study is proposed to conclude the paper.

First, when considering a functional family with fixed dimension, we discuss the confidence band derived from Sun and Loader (1994). It is biased if the dimension is not high enough to approximate well the true function. We then propose a new confidence band that corrects this bias. To do this, the bias is estimated and the additional randomness is controlled. A selection criterion is proposed to select the best dimension. Unfortunately, the two types of randomness lead to a wider confidence band, and this result is therefore no more interesting than the naive one, which consists of taking the largest possible dimension $L_{\max}$. Finally, a heuristic selection criterion is proposed to select the dimension on the first confidence band, which has not corrected the bias. It takes into account the bias as well as the variance, to select a moderate dimension. Numerical experiments show that this criterion, combined with the Legendre basis, achieves the best performance when considering the confidence level and the width of the corresponding simultaneous confidence band.

An interesting next step, but out of the scope of this paper, is a theoretical study of this criterion. No result, to our knowledge, exists for confidence bands with the supremum norm. The Euclidean norm has been extensively studied, but is not of interest here, where we want to ensure that the tube is valid as a whole. The supremum norm, on the other hand, is difficult to study theoretically. Furthermore, a key point here is the randomness of the criterion, which must also be taken into account, through an oracle inequality for example.

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

## 9 Appendix: proofs

### 9.1 Proof of Proposition 2.1

Let us prove the first point. We have

$$\mathbb{E}(\hat{\underline{\mu}}^L) = (\mathbf{B}_L^T \mathbf{B}_L)^{-1} \mathbf{B}_L^T \mathbb{E}(\mathbf{y}) = (\mathbf{B}_L^T \mathbf{B}_L)^{-1} \mathbf{B}_L^T \mathbf{B}_{L^*} \mu^{L^*} =: \underline{\mu}^L.$$

The theory of the linear model gives that the variance of $\hat{\underline{\mu}}^L$ is equal to $\sigma^2 (\mathbf{B}^T \mathbf{B})^{-1} \mathbf{B}^T \Sigma \mathbf{B} (\mathbf{B}^T \mathbf{B})^{-1}$ with $\Sigma = Diag(\Sigma_1, \dots, \Sigma_N)$ the $nN \times nN$ covariance matrix of $\mathbf{y}$. So finally, we have

$$\hat{\underline{\mu}}^L \sim \mathcal{N}\left(\underline{\mu}^L, \sigma^2 \Sigma_B^{L, L^\varepsilon}\right).$$

Now we can easily deduce the distribution of $\hat{\underline{f}}^L(t)$, for each $t \in [0, 1]$:

$$\hat{\underline{f}}^L(t) - \mathbf{f}^L(t) \sim \mathcal{N}\left(0, \sigma^2 B(t) \Sigma_B^{L, L^\varepsilon} B(t)^T\right).$$

To prove that $(\hat{\underline{f}}^L - f^L)$ is a Gaussian process, we consider any finite sequence of times $(t_1, \dots, t_d) \in [0, 1]$. The sequence $(\hat{\underline{f}}^L(t_1) - f^L(t_1), \dots, \hat{\underline{f}}^L(t_d) - f^L(t_d))$ is Gaussian, centered, and the covariance is equal to $cov(\hat{\underline{f}}^L(t_1) - f^L(t_1), \hat{\underline{f}}^L(t_2) - f^L(t_2)) = \sigma^2 B(t_1) \Sigma_B^{L, L^\varepsilon} B(t_2)^T$. Thus, the process is Gaussian.

### 9.2 Proof of Theorem 3.2

For all $t$, for all $\omega \in \Omega$,

$$\lim_{n \to +\infty} \underline{f}_n^L(t) - f^L(t) = 0$$

which means for all $\varepsilon > 0$, there exists $N_0$ such that for all $n > N_0$,

$$|\underline{f}^L(t) - f^L(t)| \le \varepsilon.$$

Then, we have, with probability $1 - \alpha$,

$$|\hat{\underline{f}}^L(t) - \underline{f}^L(t)| + |\underline{f}^L(t) - f^L(t)| \le \hat{d}^L(t) + \varepsilon$$

with $\hat{d}^L(t) = \hat{c}^L \sqrt{\hat{C}_L(t, t)/N}$ and $\hat{c}^L$ defined as the solution of Equation 4.

Then, with probability larger than $1 - \alpha$,

$$|\hat{\underline{f}}^L(t) - \underline{f}^L(t) + \underline{f}^L(t) - f^L(t)| \le \hat{d}^L(t) + \varepsilon$$

### 9.3 Proof of Proposition 4.1

To simplify the notations, let us denote $a(t) = \underline{f}^L(t) - \hat{\underline{f}}_1^L(t)$ and $b(t) = \underline{f}^{L_{max}, L^*}(t) - \underline{f}^L(t) - (\hat{\underline{f}}_2^{L_{max}, L^*}(t) - \hat{\underline{f}}_2^L(t))$. We have

$$P\left(\exists t |a(t) + b(t)| \ge \hat{d}_1^L(t) + \hat{d}_2^{L, L_{max}}(t)\right) \le P\left(\exists t |a(t)| + |b(t)| \ge \hat{d}_1^L(t) + \hat{d}_2^{L, L_{max}}(t)\right)$$
$$= P\left(\exists t |a(t)| \ge \hat{d}_1^L(t)\right) P\left(\exists t |b(t)| \ge \hat{d}_2^{L, L_{max}}(t)\right) = \alpha \beta.$$

The last equality holds thanks to the independence of the two sub-samples.

## 10   Appendix: more experiments

The properties of the coefficients are illustrated in Figure 10. The true dimension is $L^* = 11$. Three families are considered, Fourier, Legendre and Splines. The plots display the absolute difference between the coefficients $\mu_\ell^{L^*}$ and the projected coefficients $\mu^L$, for different $\ell$ in x-axis and for different values of $L$ and $n$ in the y-axis, namely a case with $L < \bar{L}^*$ and two values of $n$: $L = 7, n = 20$ and $L = 7, n = 100$; and a case with $L > L^*$ and two values of $n$: $L = 15, n = 20$ and $L = 15, n = 100$. The absolute difference is represented as a gradient of color, this gradient being adapted to each functional family. We can see that as Legendre (resp. Fourier) are orthonormal (resp. orthogonal) families, the differences are close to 0 when $L = 15$, whatever the values of $n$. When $L < L^*$, the difference is close to 0 when $n$ is large. This property does not hold for the spline family, which is not orthogonal.

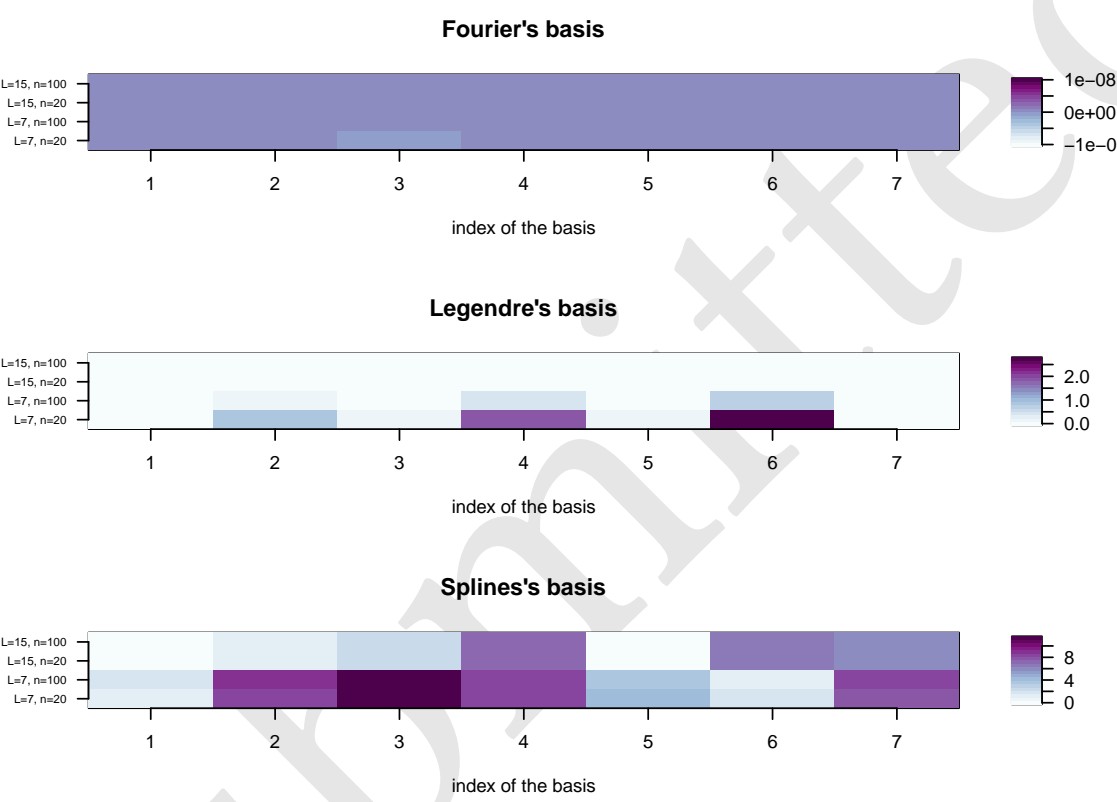

Figure 10: Illustrative example. The true dimension is 11, we generate the coefficients with three families, Fourier (which is orthogonal), Legendre (which is orthonormal) and the splines (which are not orthogonal wrt the standard scalar product). In the y-axis, two dimensions of the family (7 or 15) and two numbers of timepoints (20 or 100) are compared. We plot in x-axis the value of the absolute difference between the true coefficients and their approximations for the first 7 coefficients of the basis. The color scale is adapted to each functional basis.

