# OpenReview forum: "Should we correct the bias in Confidence Bands for Repeated Functional Data?"
_Computo — Accepted by Computo_

### Review · Reviewer_TsLK · 2026-02-07

**Summary Of Contributions:**

The paper presents a novel method to construct confidence bands for a functional model. First functional data are embedded in a finite dimensional space using a basis expansion. then confidence bands are constructed in order to take into account the fact that the dimension of the space is not known.

**Audience:**

Yes

**Broader Impact Concerns:**

no concerns

**Claims And Evidence:**

Yes

**Requested Changes:**

Major changes:
1. I suggest evaluating what happens when functional data do not lie on a finite dimensional space. This could be done for instance generating directly the data without relying on basis expansions, or at least using a very large $L*$ in data generating process (e.g., setting $L*>L_{max}$. In addition, a theoretical discussion of such cases would be very interesting.
2. Even if data truly lie in a finite dimensional space, in some cases one would not know the functional basis to use in practice. For instance, in the case study all three bases are used. It would be interesting to study numerically what happens when the basis is misspecified (e.g., data are generated with B-splines and the bands are evaluated using the Legendre basis).
3. Section 4.2: why is the infinite norm giving preference to smooth bands? Is it something that is observed in practice? Theoretically, there is no clear reason for that since the infinite norm is not affected by smoothness.

Minor changes
1. Beginning of Page 5: the quantity $t_j$ is undefined. In addition: are the locations $t_j$ assumed to be the same for all individuals, or this condition could be relaxed?
2. Page 5, line 174: is Assumption 2 truly an assumption? It is not met by for instance Splines, so it is not fully needed in the derivation of the bands.
3. Page 6 line 203: the -> then.
4. A careful revision of the English is recommended to improve clarity and readability.

**Strengths And Weaknesses:**

Strength: basis expansions are used broadly in functional data analysis, and the proposed approach can be used within any basis expansion framework. The error introduced by smoothing data with a finite dimensional basis function is taken into account when computing the confidence bands.

Weaknesses: functional data can typically be very well approximated in a finite dimensional space using a basis expansion, but might not lie in such finite dimensional space, unless $L*=\infty$. In this case Assumption 1 is not met and the approach proposed in the current paper would not be valid. In my opinion at least a discussion of such an issue, together with a numerical exploration, would improve the quality of the paper.

---

### Comment · Editors_In_Chief · 2026-06-08
**Final acceptance**

Dear authors,

Thank you for your reply to the remaining comments. Your paper is now formally accepted for publication in Computo and we are ready to enter the production step.

Can you double check the metadata of the article are correctly filled at
https://github.com/EmilieDevijver/SCB/blob/main/ConfBand.qmd#L3-L14 ?

I have just invited the corresponding author as a collaborator of computorg, the github organization of Computo. Please transfer the ownership of your repository to computorg as follows: at the bottom of https://github.com/EmilieDevijver/SCB/settings, click "Transfer ownership" and choose "computorg" in "Select one of my organizations".

Then we will be able to proceed with the publication of your article.

Thanks again submitting your work to Computo, and for your patience with the revision process.

---

### Decision · Action_Editor_yAvA · 2026-04-07

**Recommendation:** Accept with minor revision

**Comment:**

The two main remaining points to address are the behavior when functional data do not lie on a finite dimensional space (theoretical discussion and/or numerical experiments), and basis misspecification (numerical experiments).

**Audience:**

Fits well with the scope of Computo

**Claims And Evidence:**

The authors have addressed the concerns raised by the referees regarding the first submission (theoretical results, organization of the paper, numerical comparison to competitive methods).

In agreement with the elements provided by the reviewer, I consider that the claims and evidence in this submission are now sufficiently solid for publication, provided that the outstanding points raised by the referee are addressed.

---

> ### Decision · Editors_In_Chief · 2026-04-07
>
> I approve the AE's decision.

---

> ### Author Response · Authors · 2026-06-07
>
> We would like to thank the reviewer and the action editor for their constructive comments. We have updated the paper to address all the points raised. Below, we detail the major comments and our responses
>
> 1. Non-finite-dimensional functional data
>
> I suggest evaluating what happens when functional data do not lie on a finite-dimensional space. This could be done, for instance, by generating data directly without relying on basis expansions, or at least by using a very large dimension in the data-generating process (e.g., setting a high truncation parameter). In addition, a theoretical discussion of such cases would be very interesting.
> In some sense, the real data analysis (Section 7) already addresses this point, as the dataset does not lie in a finite-dimensional space.
>
> 2. Misspecified functional basis
>
> Even if data truly lie in a finite-dimensional space, in some cases one would not know the functional basis to use in practice. For instance, in the case study, all three bases are used. It would be interesting to study numerically what happens when the basis is misspecified (e.g., data are generated with B-splines and the bands are evaluated using the Legendre basis).
> In Section 6.5, there is a numerical study where data are generated using the B-spline basis and confidence bands are evaluated using both the Fourier and Legendre bases. This allows us to assess the robustness of our method to basis misspecification.
>
> 3. Infinite norm and smooth bands (Section 4.2)
>
>  Why is the infinite norm giving preference to smooth bands? Is this something that is observed in practice? Theoretically, there is no clear reason for that since the infinite norm is not affected by smoothness.
> The reviewer is correct. We have revised the text to clarify that our initial statement was an oversimplification. The preference for smooth bands is an empirical observation, not a theoretical property of the infinite norm.
>
> We have also addressed the minor comments by defining $t_j$ explicitly, removing Assumption 2, proofreading the manuscript for English language and clarity.
>
> We hope that these revisions address the outstanding points raised by the referee.